# Agenda-setting in the clinical encounter: A systematic review protocol

**Ailyn Sierpe**[1], **Renata W. Yen**[1,2], **Gabrielle Stevens**[1], **Aricca D. Van Citters**[1], **Glyn Elwyn**[1], **Catherine H. Saunders**[1,3]*

**1** The Dartmouth Institute for Health Policy and Clinical Practice, Geisel School of Medicine at Dartmouth College, Lebanon, NH, United States of America, **2** Center for Technology and Behavioral Health, Geisel School of Medicine at Dartmouth College, Lebanon, NH, United States of America, **3** Dartmouth Health, Lebanon, NH, United States of America

* catherine.hylas.saunders@dartmouth.edu

## Abstract

### Introduction

Agenda-setting is a collaborative communication strategy used by a clinician before or at the start of a clinical encounter to work together with the patient to "elicit, propose, and organize" topics to be discussed during the encounter. While clinical visit agenda-setting has been acknowledged as an important element of patient-centered communication, the effectiveness of agenda-setting interventions in improving healthcare outcomes is unclear. To our knowledge, no systematic review has examined clinical visit agenda-setting interventions.

### Methods and analysis

The primary aim of the systematic review will be to assess the effects of agenda-setting interventions on outcomes relating to the clinical encounter itself, patients, and clinicians, as well as any other study-specified outcomes. Our secondary aims will be to examine the characteristics and delivery attributes of agenda-setting interventions, as well as how agenda-setting has been operationalized and measured. We will search selected databases (APA PsycInfo, Cochrane Central Register of Controlled Trials, Cochrane Database of Systematic Reviews, Cumulative Index to Nursing and Allied Health Literature, MEDLINE via PubMed, ProQuest, Scopus, and Web of Science) and gray literature from inception until date of search. All studies comparing a clinical visit agenda-setting intervention with either usual care or another agenda-setting intervention will be included. Two independent reviewers will complete article screening and data extraction, with a third independent reviewer resolving any conflicts. We will assess all studies' methodological quality and the quality of their evidence using standardized criteria. If a sufficient number of studies report the same outcomes, we will pool their results and perform a meta-analysis of those outcomes. We will also synthesize all results qualitatively, regardless of whether we are able to complete a meta-analysis.

**Data Availability Statement:** No datasets were generated or analyzed in the preparation of this protocol. All relevant data will be made available upon study completion.

**Funding:** The author(s) received no specific funding for this work.

**Competing interests:** Ailyn Sierpe: No conflicts of interest. Renata Yen: No conflicts of interest. Gabrielle Stevens: I have co-authored a published conference abstract about the feasibility of a goal-based agenda-setting intervention (citation below). This publication may appear in search results for this review, but will not meet eligibility criteria due to the lack of a comparison group. Stevens G, Elwyn G. Feasibility of a goal-based agenda setting intervention for informing conversations in adult cystic fibrosis care: The goal talk study. Journal of Cystic Fibrosis. 2021. https://www.ScienceDirect.com/journal/journal-of-cystic-fibrosis/vol/20/suppl/S2 I have also co-authored a publication about the evaluation of a question prompt intervention to increase shared-decision making in clinical encounters (citation below). This publication may appear in search results for this review, but will likely not meet eligibility criteria due to the study focus. Thompson R, Stevens G, Manski R, Donnelly KZ, Agusti D, Li Z, et al. Right For Me: a pragmatic multi-arm cluster randomised controlled trial of two interventions for increasing shared decision-making about contraceptive methods. medRxiv. 2021. https://www.medrxiv.org/content/10.1101/2021.06.25.21257891v1 Additionally, I have a broad research interest in the development and evaluation of agenda-setting interventions, have received funding for these purposes in the past, and may again in the future. Aricca Van Citters: I have co-authored several publications that include agenda-setting as a component of a larger intervention (citations below), and am conducting research on the types of concerns that agenda-setting questions elicit and their impact on shared decision-making. These publications may appear in search results for this review, but will likely not meet eligibility criteria due to the study focus. Van Citters AD, Gifford AH, Brady C, Dunitz JM, Elmhirst M, Flath J, Laguna TA, Moore B, Prickett ML, Riordan M, Savant AP, Gore W, Jian S, Soper M, Marshall BC, Nelson EC, Sabadosa KA. Formative evaluation of a dashboard to support coproduction of healthcare services in cystic fibrosis. J Cyst Fibros. 2020;19(5):768-76. Epub 2020/05/02. doi: 10.1016/j.jcf.2020.03.009. PubMed PMID: 32354650. Van Citters AD, Holthoff MM, Kennedy AM, Melmed GY, Oberai R, Siegel CA, Weaver A, Nelson EC. Point-of-care dashboards promote coproduction of healthcare services for patients with inflammatory bowel disease. Int J Qual Health Care. 2021;33 (Supplement_2):ii40-ii7. doi: 10.1093/intqhc/mzab067. PubMed PMID: 34849970. Van Citters AD, Taxter AJ, Mathew SD, Lawson E, Eseddi J,

## PROSPERO registration number

CRD42023468045

## Introduction

Agenda-setting is a collaborative communication strategy used by clinicians before or at the start of a clinical encounter [1, 2]. It has been defined as a practice in which the clinician works together with the patient to "elicit, propose, and organize" the topics to be discussed during the encounter [1, 3, 4]. Clinical visit agenda-setting is an important element of patient-centered communication [5], and has been associated with positive outcomes such as greater patient satisfaction, increased likelihood of patients adhering to treatment recommendations, and clinicians' improved understanding of patient concerns [1, 3, 6–8]. Agenda-setting has also been shown to reduce the overall number of concerns that go unaddressed by the end of the visit, or that are not sufficiently resolved due to the patient raising them at the last minute [1, 9].

Although agenda-setting has been acknowledged as a valuable practice and is commonly included in medical education, many clinicians have a limited understanding of agenda-setting or may not perform it effectively, given the competing priorities and interests within encounters [1, 3, 4, 10–12]. Naturalistic studies of clinical visit agenda-setting have also observed considerable variation in its real-world prevalence [1, 8, 10, 13]. This is compounded by the challenge of measuring agenda-setting, which has been inconsistently defined and operationalized, without widespread adoption of any single standardized measuring tool [1, 4, 14]. For example, while some studies consider all instances of clinicians starting an encounter by asking an open-ended question to be agenda-setting [15], others have placed specific time limits for when agenda-setting must occur, such as within the first five minutes [13].

Studies of clinical visit agenda-setting have also adopted a wide range of approaches in designing interventions to increase or enhance agenda-setting. Namely, patient-facing interventions have largely centered on agenda-setting checklists or other tools to be completed by the patient or their care partner before or during the encounter, with some evidence of positive outcomes [2, 5, 16–18]. Meanwhile, clinician-facing interventions have typically involved training programs in agenda-setting skills, such as topic elicitation workshops [3, 16, 18–22]. Other studies have combined both of these approaches [18].

Along with the diverse nature of the interventions themselves, the outcomes examined by these studies have also varied substantially. For instance, some have focused on the overall duration of the visit and the number of topics discussed [5, 23], or more specifically, the number of 'surprise' topics raised only at the end of the encounter [1, 18]. As a broader measure of patient-centered communication, another study examined verbal activity, or the proportion of statements contributed by each speaker during the encounter [5]. Finally, other studies have used patient-reported measures of clinicians' communication skills and general satisfaction [3, 18]. Such variation has contributed to the challenge of determining the overall effectiveness of existing clinical visit agenda-setting interventions.

Despite the relevance and value of clinical visit agenda-setting in patient-centered communication, there is currently a gap in our understanding of the best approaches to increasing the consistent adoption and quality of agenda-setting practices. We are also not aware of any previous comprehensive reviews of the evidence for clinical visit agenda-setting interventions. To address these gaps, our systematic review will primarily aim to assess the effects of agenda-setting interventions on outcomes relating to the clinical encounter itself, patients, and clinicians,

Del Gaizo V, Ahmad J, Bajaj P, Courtnay S, Davila L, Donaldson B, Kimura Y, Lee T, Mecchella JN, Nelson EC, Pompa S, Tabussi D, Johnson LC. Enhancing Care Partnerships Using a Rheumatology Dashboard: Bringing Together What Matters Most to Both Patients and Clinicians. ACR Open Rheumatol. 2023. Epub 2023/03/01. doi: 10.1002/acr2.11533. PubMed PMID: 36852527. I have also received grant funding to test the feasibility of agenda-setting interventions, as components of larger studies. Glyn Elwyn: I have co-authored a published conference abstract about the feasibility of a goal-based agenda-setting intervention (citation below). This publication may appear in search results for this review, but will not meet eligibility criteria due to the lack of a comparison group. Stevens G, Elwyn G. Feasibility of a goal-based agenda setting intervention for informing conversations in adult cystic fibrosis care: The goal talk study. Journal of Cystic Fibrosis. 2021. https://www.ScienceDirect.com/journal/journal-of-cystic-fibrosis/vol/20/suppl/S2 I have also co-authored a publication about the evaluation of a question prompt intervention to increase shared-decision making in clinical encounters (citation below). This publication may appear in search results for this review, but will likely not meet eligibility criteria due to the study focus. Thompson R, Stevens G, Manski R, Donnelly KZ, Agusti D, Li Z, et al. Right For Me: a pragmatic multi-arm cluster randomised controlled trial of two interventions for increasing shared decision-making about contraceptive methods. medRxiv. 2021. https://www.medrxiv.org/content/10.1101/2021.06.25.21257891v1 I have also co-authored a publication about the development of a clinical visit agenda-setting tool, called Serious Illness Topics (citation below). This publication may appear in search results for this review, but will likely not meet eligibility criteria due to study design. Saunders CH, Durand M-A, Scalia P, Kirkland KB, MacMartin MA, Barnato AE, et al. 'It helps us say what's important…' Developing Serious Illness Topics: A clinical visit agenda-setting tool. Patient Educ Couns 2023;113:107764. https://doi.org/10.1016/j.pec.2023.107764. Additionally, I have a broad research interest in the development and evaluation of agenda-setting interventions, have received funding for these purposes in the past, and may again in the future. Catherine Saunders: I have co-authored a publication about the development of a clinical visit agenda-setting tool, called Serious Illness Topics (citation below). This publication may appear in search results for this review, but will likely not meet eligibility criteria due to study design. Saunders CH, Durand M-A, Scalia P, Kirkland KB, MacMartin MA, Barnato AE, et al. 'It

as well as any other study-specified outcomes. Our secondary aims will be to examine the characteristics and delivery attributes of agenda-setting interventions, as well as how agenda-setting has been operationalized and measured.

## Methods and analysis

We will conduct a systematic review of studies of clinical visit agenda-setting interventions. In response to our primary and secondary aims, we will address the following research questions:

1. What are the effects of clinical visit agenda-setting interventions on outcomes relating to the clinical encounter itself, patients, and clinicians, as well as any other study-specified outcomes?

2. What are the characteristics and delivery attributes of clinical visit agenda-setting interventions?

3. How has clinical visit agenda-setting been operationalized and measured?

We have registered this protocol with PROSPERO (CRD42023468045). We developed our methods with guidance from the Cochrane Handbook for Systematic Reviews of Interventions [24] and the Preferred Reporting Items for Systematic Review and Meta-Analysis Protocols (PRISMA-P) checklist (see S1 File) [25]. After completing the review, we will report our methods and results using the full PRISMA guidelines [26].

### Preliminary searches

We conducted preliminary searches in PROSPERO, the Cochrane Database of Systematic Reviews, and Google Scholar to identify any existing systematic reviews and to assess the quantity of articles that could potentially be included. We used the keywords "agenda-setting," "agenda-mapping," "topic elicitation," "topic solicitation," "patient agenda," "patient concerns," "patient priorities," "opening statement," "visit opening," and "visit structure." We assessed the most cited literature reviews and randomized control trials identified from the searches. We did not find any systematic reviews of clinical visit agenda-setting.

### Primary search strategy

We developed and piloted our search strategy (see S2 File) with the assistance of two biomedical research librarians at Dartmouth College. We will perform electronic searches in APA PsycInfo, Cochrane Central Register of Controlled Trials, Cochrane Database of Systematic Reviews, Cumulative Index to Nursing and Allied Health Literature, MEDLINE via PubMed, ProQuest, Scopus, and Web of Science. With the broad nature of clinical visit agenda-setting, it is difficult to determine a specific start date for when interventions would have first been developed. We will therefore search for articles from database inception until date of search (see Table 1). As we did not identify any relevant MeSH terms referring to clinical visit agenda-setting, we will use sets of keywords instead. We will not use limits or language restrictions. A list of all searches performed will be included as a supplement to the final report.

### Secondary search strategy

To reduce potential selection bias, we will ensure an exhaustive search by supplementing our primary search strategy with the following secondary search methods (see S2 File):

- Searching the reference lists of all included articles to find any articles not identified in our initial searches (backward citation search).

**Table 1. Databases in primary search strategy.**

| Database name | Years covered |
| --- | --- |
| APA PsycInfo | 1806–present |
| The Cochrane Library (Central Register of Controlled Trials) | 1989–present |
| The Cochrane Library (Cochrane Database of Systematic Reviews) | 1996–present |
| Cumulative Index to Nursing and Allied Health Literature | 1981–present |
| MEDLINE via PubMed | 1946–present |
| ProQuest | 1861–present |
| Scopus | 1788–present |
| Web of Science | 1900–present |

- Searching the citations of all included articles by using the 'cited by' feature in Web of Science (forward citation search).

- Searching Google Scholar and screening the first 25 pages of results. If results continue to be relevant after 25 pages, we will continue reviewing additional pages until two consecutive pages yield no relevant articles.

- Searching ClinicalTrials.gov for registered clinical trials.

- Searching key journals, relevant academic conference proceedings, and gray literature such as technical reports and works in progress.

- Searching reference lists provided by colleagues with expertise in research on clinical visit agenda-setting.

## Inclusion criteria

**Types of studies.** We will include randomized studies of clinical visit agenda-setting interventions where a comparison is made to either no intervention (usual care) or to at least one other agenda-setting intervention (see S3 File). We will also include non-randomized studies, provided there is a comparison strategy (pre-post or quasi-experimental designs). To allow us to draw stronger conclusions on intervention effectiveness, we will exclude non-comparative designs and qualitative studies. As we anticipate there being relatively little research on this topic, based on our preliminary searches and knowledge of the literature in this field, we will include feasibility studies and pilot trials. We will also not impose restrictions on publication language. A colleague fluent in the target language will translate any non-English articles. Where this is not possible, we will use Google Translate and ask the authors to confirm the appropriateness of our interpretations.

**Types of participants.** We will include all participants involved in a clinical visit agenda-setting intervention, including patients, their care partners, clinicians, and any other groups. As agenda-setting can be practiced with any group or condition and in any clinical environment, we will include all clinical settings and participant groups.

**Types of interventions.** We will include all interventions that specifically aim to promote or increase clinical visit agenda-setting. The interventions may be patient-facing, clinician-facing, or both. We will exclude any interventions that only incidentally involve agenda-setting, such as those primarily focused on goal-setting. We have developed a working definition of agenda-setting for the purposes of this review by combining the characteristics of agenda-setting commonly described in the literature (see S3 File). We define agenda-setting as a practice

in which a clinician works collaboratively with a patient to elicit, and often propose or organize, the topics to be discussed during a clinical encounter [1, 3]. Some of the literature aiming to define and characterize agenda-setting has emphasized the need for it to be performed to exhaustion. This means the clinician continues to elicit additional topics until none remain [2, 4, 14, 27, 28]. We argue exhaustion is not a practical goal given the time limitations of clinical visits, so we instead focus on the general elicitation of topics. We acknowledge that exhaustion may be appropriate for assessing the quality of agenda-setting, i.e. scoring agenda-setting quality in encounters [14]. Additionally, while some researchers have understood agenda-setting as taking place at any time during a visit [4, 27], both observational studies and studies of agenda-setting interventions have acknowledged the positioning of agenda-setting before or early on in the encounter as best practice [1, 2, 8, 9, 13, 15, 29]. We will therefore only include interventions in which the agenda-setting occurs before or at the start of the encounter, rather than the end.

**Types of outcomes.** To address our primary aim, we will include all clinical and non-clinical outcomes relating to the clinical encounters themselves, patients, or clinicians. As our search strategy piloting indicated a relatively small volume of literature (see S2 File), we will also include any other study-specified primary and secondary outcomes. Based on our preliminary searches, this may include but will not be limited to visit process outcomes, patient or care partner-reported outcomes, clinician-reported outcomes, and observer measures. To address our secondary aims, we will include outcomes relating to any agenda-setting measurement instruments, such as their reliability or validity. With the highly diverse nature of our outcomes of interest, we will include all units of measurements used in the included studies.

## Initial screening

We will upload all articles retrieved from the searches to Rayyan [30] or a similar systematic review management software. We will then remove any duplicate items. The data will be continuously backed up and managed by the software.

Two independent reviewers will first pilot our preliminary study inclusion criteria (see S3 File) with approximately 50–100 article abstracts. After confirming their shared understanding of the inclusion criteria and revising as needed, the reviewers will begin screening by reviewing the title and abstract of each article to determine whether it meets the final criteria. To maintain consistency, we will use a standardized list of study inclusion criteria with detailed descriptions of each point. Articles that pass this first round of screening will then advance onto full-text review. The two reviewers will again independently screen for articles matching the inclusion criteria. Any disagreements at either stage of screening will be resolved by discussion with a third independent reviewer.

## Data extraction

We will extract data from the articles meeting our study inclusion criteria using a predesigned standardized data collection form. The form will consist of a spreadsheet, with the rows listing the articles and the columns indicating the data points to be extracted. To extract information on intervention characteristics, we will include items adapted from the TIDieR (Template for Intervention Description and Replication) checklist in the data collection form [31]. We will extract details about the 1) authors, 2) publication year, 3) country where the study took place, 4) study design, 5) study aims and research questions, 6) participant characteristics and sample size, 7) study setting, 8) characteristics and duration of the intervention, 9) follow-up, 10) comparison conditions, 11) number of participants included in the analysis for both intervention

and comparison groups, 12) measurement of clinical visit agenda-setting, and 13) all primary and secondary outcomes.

Two independent reviewers will first pilot the form using a small set of articles. After revising the form as needed, the reviewers will progress to full data extraction for the included studies, with disagreements again resolved by discussion with a third independent reviewer. We will attempt to obtain any missing data by contacting article authors. If we are unable to reach them, we will compute remaining values where appropriate [32, 33].

## Methodological quality assessment

As our inclusion criteria cover multiple study designs, we will use two different risk-of-bias tools to evaluate the included studies: RoB 2 (the Cochrane risk-of-bias tool) for randomized control trials [34], and ROBINS-I (Risk Of Bias In Non-randomised Studies - of Interventions) for non-randomized trials [35]. All reviewers will be trained on using the tools before beginning their assessments. Two independent reviewers will assign each study a rating of methodological quality using the risk-of-bias tool corresponding to its design. A third independent reviewer will resolve any disagreements. We will also use GRADE (Grading of Recommendations, Assessment, Development and Evaluation) to evaluate the certainty of the evidence reported [36, 37]. We will use the RoB 2 and ROBINS-I results and other components of the included studies to apply the GRADE scoring. We will use this same approach to assessing methodological quality for both peer-reviewed articles and gray literature.

## Data synthesis

All included studies, regardless of their source, will be synthesized using the same approach. For any outcomes reported by at least two studies, we will pool the results and conduct a meta-analysis using statistical software [38, 39]. We will present dichotomous data as relative risks and continuous data as means or standardized mean differences, with all analyses using 95% confidence intervals. As it is unlikely that the intervention effects will be identical, we will use a random effects model, assuming significance at $p<0.05$ [39, 40]. For quantitatively pooled results, we will assess heterogeneity using $\chi^2$ and $I^2$ tests. For outcomes not included in the meta-analysis, we will qualitatively assess heterogeneity by visually inspecting the results of the individual studies to identify any notable outliers. If there is more than minimal heterogeneity ($p<0.10$ and $I^2>40\%$) [39, 41], we will identify the sources of heterogeneity and assess their impact on the meta-analysis. For outcomes reported by at least 10 studies, we will assess reporting bias using funnel plots, where effect estimates of the common outcome measure are plotted against trial sample size. We will then conduct Egger's regressions to statistically assess for asymmetry, assuming significance at $p<0.05$. For outcomes reported by fewer than 10 studies, we will assess reporting bias qualitatively [42].

We will also produce a narrative synthesis of the included studies [43]. The data from each study, including intervention characteristics, setting, participant characteristics, and enrollment metrics, will be used to build evidence tables according to the SWiM (Synthesis Without Meta-analysis) reporting guidelines [44]. Using the evidence tables, we will qualitatively summarize the direction and intensity of the effects of the agenda-setting interventions. We will also summarize results relating to our secondary aims. To increase the validity of our narrative synthesis, we will adopt the participatory approach outlined in Practical Thematic Analysis [45] and will include a clinician and a patient partner as members of the research team. Our partners will take part in interpreting and summarizing the results of the included studies, as well as provide feedback on the initial results of the systematic review.

We will perform sensitivity analyses by removing the methodologically weakest studies, outliers in participant or intervention characteristics, and any studies where we computed missing values to determine whether the intervention effects change or remain the same. We will also perform subgroup analyses, where there is sufficient data. While we will not predetermine specific subgroups, this would ideally include separate analyses of patient-facing interventions, clinician-facing interventions, and those combining both approaches. If the data is sufficient, we also aim to perform analyses by the included studies' settings and participant characteristics.

## Ethics and dissemination

The results of this systematic review will be submitted for publication in a peer-reviewed journal and presented at relevant conferences. We will report on the full review in accordance with the PRISMA (Preferred Reporting Items for Systematic Reviews and Meta-Analyses) statement [26]. In the event of any amendments to the protocol, the date of each amendment will be accompanied by a description of the change and its rationale. Changes will not be incorporated into the original protocol.

## Discussion

Clinical visit agenda-setting has been repeatedly recognized as a best practice in patient-centered communication, with its ability to foster shared understanding and engagement between patients and clinicians in organizing and addressing topics of importance [4, 27]. However, recent signals have pointed to substantial variation in how often agenda-setting actually occurs [1, 13], clinicians' ability to perform high-quality agenda-setting [10, 12], and the specific benefits of agenda-setting for different patient populations and clinical contexts [1, 3, 9]. Despite this, there has not yet been a comprehensive evaluation of interventions to increase or improve agenda-setting in the clinical encounter. To address this gap, this systematic review will evaluate the effectiveness of clinical visit agenda-setting interventions in improving outcomes for the encounters themselves, patients, and clinicians. Such a review has been long overdue, particularly given the broader value of agenda-setting as a path towards better aligning healthcare delivery with patients' individual needs [6, 7]. By contributing to evidence on how to best promote clinical visit agenda-setting, we can inform ongoing efforts to develop, test, and implement effective targeted interventions that will serve to advance patient-centered communication as a whole.

## Supporting information

**S1 File. PRISMA-P checklist.**
(DOCX)

**S2 File. Piloted search strategy.**
(DOCX)

**S3 File. Summary of inclusion criteria and agenda-setting definitions.**
(DOCX)

## Acknowledgments

We thank Heather Blunt and Elaina Vitale for their assistance in developing the search strategy.

## Author Contributions

**Conceptualization:** Ailyn Sierpe, Catherine H. Saunders.

**Methodology:** Ailyn Sierpe, Renata W. Yen, Gabrielle Stevens, Glyn Elwyn, Catherine H. Saunders.

**Project administration:** Ailyn Sierpe.

**Supervision:** Glyn Elwyn, Catherine H. Saunders.

**Writing – original draft:** Ailyn Sierpe, Renata W. Yen, Gabrielle Stevens, Catherine H. Saunders.

**Writing – review & editing:** Ailyn Sierpe, Renata W. Yen, Gabrielle Stevens, Aricca D. Van Citters, Glyn Elwyn, Catherine H. Saunders.

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
