## [Decision Letter · Decision Letter 0]

18 Jun 2024

PONE-D-24-12064Agenda-setting in the clinical encounter: a systematic review protocolPLOS ONE

Dear Dr. Saunders,

Thank you for submitting your manuscript to PLOS ONE. After careful consideration, we feel that it has merit but does not fully meet PLOS ONE’s publication criteria as it currently stands. Therefore, we invite you to submit a revised version of the manuscript that addresses the points raised during the review process.

We look forward to receiving your revised manuscript.

Kind regards,

Yohannes Kebede, Ph.D.

Guest Editor

PLOS ONE

Journal Requirements:

Reviewers' comments:

Reviewer's Responses to Questions

Comments to the Author

1. Does the manuscript provide a valid rationale for the proposed study, with clearly identified and justified research questions?

Reviewer #1: Partly

Reviewer #2: Yes

2. Is the protocol technically sound and planned in a manner that will lead to a meaningful outcome and allow testing the stated hypotheses?

Reviewer #1: Partly

Reviewer #2: Yes

3. Is the methodology feasible and described in sufficient detail to allow the work to be replicable?

Reviewer #1: Yes

Reviewer #2: Yes

4. Have the authors described where all data underlying the findings will be made available when the study is complete?

Reviewer #1: Yes

Reviewer #2: Yes

5. Is the manuscript presented in an intelligible fashion and written in standard English?

Reviewer #1: No

Reviewer #2: Yes

6. Review Comments to the Author

You may also provide optional suggestions and comments to authors that they might find helpful in planning their study.

**Reviewer #1:** 1. Have you planned to perform sensitivity analyses to explore the robustness of the findings?

2. How you are going to pool the findings, it should be clearly described?

3. Are there enough articles to do systematic review and metaanalysis?

4. Study period needs justification, as you planned to start from 1788?

5. How you are going to determine sufficient number of studies report the same outcomes, to pool their results?

6. The review question should precise and overall writing issues should be revised

7. Better to provide a thorough and detailed explanation of the interventions or exposures being examined, including precise descriptions that would enable others to replicate the study or determine its relevance to different contexts.

8. Specific techniques to synthesize the data should be precisely described?

9. What is standardized data collection form, how, it should be clear.

**Reviewer #2: **The comment is provided in track change. You just need to refine those like: The list number of studies to be considered to conduct meta analysis. The planned sensitivity and subgroup analysis is limited so broaden to setting, and population group.

7. PLOS authors have the option to publish the peer review history of their article (what does this mean?). If published, this will include your full peer review and any attached files.

Do you want your identity to be public for this peer review? For information about this choice, including consent withdrawal, please see our Privacy Policy.

Reviewer #1: No

Reviewer #2: Yes: Sabit Ababor Ababulgu

---

## [Author Response · Author response to Decision Letter 0]

14 Aug 2024

Please see our response to reviewers in the attached document.

---

## [Decision Letter · Decision Letter 1]

22 Aug 2024

PONE-D-24-12064R1Agenda-setting in the clinical encounter: A systematic review protocolPLOS ONE

Dear Dr. Saunders,

Thank you for submitting your manuscript to PLOS ONE. After careful consideration, we feel that it has merit but does not fully meet PLOS ONE’s publication criteria as it currently stands. Therefore, we invite you to submit a revised version of the manuscript that addresses the points raised during the review process. Please submit your revised manuscript by Oct 06 2024 11:59PM. If you will need more time than this to complete your revisions, please reply to this message or contact the journal office at plosone@plos.org. Please include the following items when submitting your revised manuscript:A rebuttal letter that responds to each point raised by the academic editor and reviewer(s). You should upload this letter as a separate file labeled 'Response to Reviewers'.A marked-up copy of your manuscript that highlights changes made to the original version. You should upload this as a separate file labeled 'Revised Manuscript with Track Changes'.An unmarked version of your revised paper without tracked changes. You should upload this as a separate file labeled 'Manuscript'.If applicable, we recommend that you deposit your laboratory protocols in protocols.io to enhance the reproducibility of your results. Protocols.io assigns your protocol its own identifier (DOI) so that it can be cited independently in the future. For instructions see: https://journals.plos.org/plosone/s/submission-guidelines#loc-laboratory-protocols. Additionally, PLOS ONE offers an option for publishing peer-reviewed Lab Protocol articles, which describe protocols hosted on protocols.io. Read more information on sharing protocols at https://plos.org/protocols?utm_medium=editorial-email&utm_source=authorletters&utm_campaign=protocols.

We look forward to receiving your revised manuscript.

Kind regards,

Yohannes Kebede, Ph.D.

Guest Editor

PLOS ONE

Journal Requirements:

Reviewers' comments:

Reviewer's Responses to Questions

**Comments to the Author**

1. Does the manuscript provide a valid rationale for the proposed study, with clearly identified and justified research questions?

Reviewer #1: Yes

Reviewer #2: Yes

2. Is the protocol technically sound and planned in a manner that will lead to a meaningful outcome and allow testing the stated hypotheses?

Reviewer #1: Yes

Reviewer #2: Yes

3. Is the methodology feasible and described in sufficient detail to allow the work to be replicable?

Reviewer #1: Yes

Reviewer #2: Yes

4. Have the authors described where all data underlying the findings will be made available when the study is complete?

Reviewer #1: Yes

Reviewer #2: Yes

5. Is the manuscript presented in an intelligible fashion and written in standard English?

Reviewer #1: Yes

Reviewer #2: Yes

6. Review Comments to the Author

You may also provide optional suggestions and comments to authors that they might find helpful in planning their study.

Reviewer #1: My comments were addressed. Here is additional suggestions for improvement.

Please discuss potential biases in the included studies, such as selection and publication bias, and outline your approach to addressing these in the review. Additionally, clarify how you will evaluate gray literature alongside peer-reviewed evidence, including your assessment of their reliability. It would be beneficial to express your confidence even in peer-reviewed studies and how you plan to integrate both types of literature in your analysis.

I suggest elaborating on the specific qualitative synthesis methods you plan to employ. Providing details on how you will analyze and interpret the qualitative findings, as well as how you will ensure rigor and credibility, will strengthen the protocol.

Reviewer #2: The research questions to be addressed is clear and able to add value to the knowledge base. The effectiveness of

agenda-setting interventions in improving healthcare outcomes is not clear and I couldn't get a systematic review that has examined clinical visit agenda-setting interventions. The protocol is technically and methodologically sound to conduct the stated systematic review. The the search, study selection and extraction methods describes ensures the feasibility to replicate. So, I suggest accepting this protocol to be published.

7. PLOS authors have the option to publish the peer review history of their article (what does this mean?). If published, this will include your full peer review and any attached files.

Reviewer #1: **Yes: **Daba Abdissa

Reviewer #2: **Yes: **Sabit Ababor Ababulgu

---

## [Editor Report · Decision Letter 2]

10 Oct 2024

Agenda-setting in the clinical encounter: A systematic review protocol

PONE-D-24-12064R2

Dear Dr. Saunders,

We’re pleased to inform you that your manuscript has been judged scientifically suitable for publication and will be formally accepted for publication once it meets all outstanding technical requirements.

Kind regards,

Yohannes Kebede, Ph.D.

Guest Editor

PLOS ONE
---

## [Editor Report · Acceptance letter]

14 Oct 2024

PONE-D-24-12064R2 

PLOS ONE

Dear Dr. Saunders, 

I'm pleased to inform you that your manuscript has been deemed suitable for publication in PLOS ONE. Congratulations! Your manuscript is now being handed over to our production team.

Kind regards, 

on behalf of

Dr. Yohannes Kebede 

Guest Editor

PLOS ONE